METHODS

# An allele-sharing, moment-based estimator of global, population-specific and population-pair $F_{ST}$ under a general model of population structure

**Jerome Goudet** [1,2]*, **Bruce S. Weir** [3]

**1** Dept Ecology & Evolution, University of Lausanne, Lausanne, Switzerland, **2** Swiss Institute of BioInformatics, University of Lausanne, Lausanne, Switzerland, **3** Department of Biostatistics, University of Washington, Seattle, Washington, United States of America

* jerome.goudet@unil.ch

**Data Availability Statement:** The data we used are publicly available (1000 Genomes project, https://ftp.1000genomes.ebi.ac.uk/vol1/ftp/release/20130502/) or simulated and code for simulations

## Abstract

Being able to properly quantify genetic differentiation is key to understanding the evolutionary potential of a species. One central parameter in this context is $F_{ST}$, the mean coancestry within populations relative to the mean coancestry between populations. Researchers have been estimating $F_{ST}$ globally or between pairs of populations for a long time. More recently, it has been proposed to estimate population-specific $F_{ST}$ values, and population-pair mean relative coancestry. Here, we review the several definitions and estimation methods of $F_{ST}$, and stress that they provide values relative to a reference population. We show the good statistical properties of an allele-sharing, method of moments based estimator of $F_{ST}$ (global, population-specific and population-pair) under a very general model of population structure. We point to the limitation of existing likelihood and Bayesian estimators when the populations are not independent. Last, we show that recent attempts to estimate absolute, rather than relative, mean coancestry fail to do so.

## Author summary

We present a general model for the evolution of coancestries within and among populations possibly of different sizes and connected by migration, allowing for any type of mating pattern. We describe a moment-based, allele-sharing estimator of global, population-specific and population-pair $F_{ST}$, the coancestries within or between pairs of populations relative to the average mean coancestry between populations and show its good statistical properties using both simulated and real, published data from the 1,000 genomes project. As few as 10 individuals per population with $10^4$ independent SNPs are sufficient to obtain accurate estimates. We evaluate statistically how our estimate compares to others, and discuss how population-pair (as opposed to pairwise) $F_{ST}$ could become a useful metric in molecular ecology and conservation genetics.

is available from https://github.com/jgx65/PlosGenetPopulationFST.

**Funding:** JG acknowledges grant 31003A-138180 of the Swiss National Science Foundation for Research support. BSW acknowledges grant GM075091 from the NIH for Research Support. The funders had no role in study design, data collection and analysis, decision to publish, or preparation of the manuscript.

**Competing interests:** The authors have declared that no competing interests exist.

## Introduction

Most species are spatially discontinuous. The partial isolation created by the discontinuity of the landscape limits gene flow. With time, genetic drift and possibly other random and non-random evolutionary processes will increase genetic differentiation among the partially isolated groups, which may favor or impede adaptation. Being able to quantify genetic differentiation properly is thus key to understanding the evolutionary potential of a species. Starting with the seminal work of Wright [1] and Malécot [2], population geneticists have been developing methods to this end.

$F_{ST}$, a quantity very commonly estimated to measure genetic differentiation, is a key parameter, but there are multiple ways in which it can be defined. For instance, $F_{ST}$ can be defined as the (intraclass) correlation of allelic indicators between gametes chosen randomly from within the same population (the mean within-populations coancestry) relative to the correlation for gametes chosen randomly from different populations (the mean between-population coancestry) [3–6]. This amounts to using the studied set of populations as a reference point. Another definition uses the mean coancestry in an ancestral population as a reference point [4, 7, 8], which allows interpretation of $F_{ST}$ in terms of identity by descent (IBD), but is valid only in very specific population models [4].

Instead of using the mean coancestry in either the current or the ancestral population, Ochoa & Storey [9] use the minimum in the current population as a proxy for the mean coancestry in the ancestral population, and we return to this later.

Several global estimators of $F_{ST}$ have been developed, using the method of moments, maximum likelihood or Bayesian methods, as reviewed in [6]. These estimators have been used to quantify gene flow among populations [10], detect selection at specific loci [11, 12] or infer whether cooperation could evolve [13].

More recently, interest has turned to population and population-pair specific $F_{ST}$ because of the realization that each population has a different history that can not be captured with a single average quantity [9, 14–20]. In terms of coancestry, for example, genetic drift would make it likely that the smallest of a set of populations would have the largest coancestry among its members, and so have the largest Fst value.

The idea of population-specific $F_{ST}$ values was first proposed by Balding & Nichols [21] in a forensic context and was followed by a discussion of Bayesian estimation methods [22]. The authors used sample allele frequencies in a forensic database to produce posterior distributions of $F_{ST}$ values for specific populations. They pointed to the greater benefit of using more loci, reducing the effects of genetic sampling variation, than of sampling more individuals and reducing the effects of statistical sampling.

Weir & Hill [14] provided a method of moment estimators for population-specific and population-pair $F_{ST}$ values in work that made explicit mention of the dependence among populations of allele frequencies. They showed that their estimators were for within-population coancestry relative to average coancestry between all pairs of populations. Population-specific estimates revealed differences at the LCT gene in humans, consistent with selection at that locus, that were not as evident with population-average estimates [23].

Beaumont & Balding [24] and Foll & Gaggiotti [25] also pointed to the use of population-specific $F_{ST}$ in the context of identifying selected loci. Gaggiotti & Foll [26] reviewed the properties of a Bayesian estimator of population-specific $F_{ST}$ under the $F$ model, which assumes all populations, possibly of different sizes, to descend from a single ancestral population and to receive immigrants from a common gene pool: a continent-islands model. These papers estimated allele frequencies in the ancestral population.

One limitation of estimators based on the $F$ model is the assumption that the populations are independent [26]. Weir & Hill [14] relaxed that assumption, as did [15–17]. Karhunen &

Ovaskainen [17] proposed a Bayesian estimator which allowed for specific migration rates between pairs of populations using an Admixture $F$ model (AFM). Neither Weir & Hill [14] nor Karhunen & Ovaskainen [17] allowed for non-random mating within populations, or provided predicted values in terms of demographic parameters. Mary-Huard & Balding [19] built on Ochoa & Storey [9] to develop a fast approximate method assuming a tree-like structure for the populations, but still assuming random mating (independence of gametes) within populations.

Weir & Goudet [18] provided estimators allowing for non-random mating within populations, and gave predicted values of mean relative coancestries for a two-population system. They also advocated, in accordance with the findings of Bhatia *et al.* [27], giving equal weight to all samples, independently of their sizes, as opposed to the estimators described in Weir & Hill [14] where samples are weighted by their respective sizes.

Most of this cited work did not investigate in depth the statistical properties of estimators. A notable exception is Gaggiotti & Foll [26] for population-specific (but not population-pair) $F_{ST}$ that showed good qualitative properties of the Bayesian estimator of population-specific $F_{ST}$ under the $F$ model, even with some departure from the model assumption of population independence. They did not provide predicted values for population-specific $F_{ST}$ other than for the island model.

Our goals here are fourfold. First we reiterate the well-known relation between $F_{ST}$ and mean coancestries [3–5, 14, 18]. Second, we derive predicted values for coancestries and $F_{ST}$ in a general model of population structure. Third, using computer simulations, we investigate the statistical properties of the moment-based estimator of population-specific and between-population $F_{ST}$ proposed by Weir & Goudet [18] using varying numbers of loci and individuals. Quantifying the statistical properties of the $F_{ST}$ estimator proposed in [18] is all the more important since Ochoa & Storey have recently claimed in this journal [9] poor properties of this estimator. We also evaluate the statistical properties of the Bayesian estimator of $F_{ST}$ put forth by [17]. Fourth, we present estimates for data from the 1,000 Genomes project [28].

## Description of the method

### Definition of $F_{ST}$

Rousset [4] gives a general definition of a parameter $F$ as

$$F \equiv \frac{Q_w - Q_b}{1 - Q_b} \tag{1}$$

with $Q_w$ and $Q_b$ respectively defined as the probability of identity, or coancestry, within and between structural units.

He then states (p 372):

"The well-known 'F-statistics' originally considered by Wright may be defined as above. [...] For Wright's $F_{ST}$, $Q_w$ is the probability of identity within a deme and $Q_b$ is the probability of identity between demes. Likewise, Wright's $F_{IS}$, $Q_w$ is the probability of identity of the two homologous genes in a diploid individual, and $Q_b$ is the probability of identity of two genes in different individuals."

This general definition shows that $F_{ST}$ and related quantities are always defined *relative to* a reference and that probabilities of identity themselves cannot be estimated. In what follows, we adopt the same framework.

Slatkin [29] uses the relation between coalescence time and probability of identity to show $F_{ST}$ could also be defined as one minus the ratio of coalescence times within and over all populations in the limit of low mutation (see also [30]). Rousset [31] reformulated it as one minus the ratio of coalescence times within and between populations in order to be consistent with Eq 1.

Bhatia *et al.* [27] review definitions of $F_{ST}$ and point out (p. 1515) that

"Wright [1] defined $F_{ST}$ as the correlation of randomly drawn gametes from the same population, relative to the total population. However, he did not clearly specify "the total population", leaving subsequent investigators to interpret its meaning."

There have been two interpretations of the "total population", either the current total population, or the ancestral one. Rousset [4, 5] points out that using the ancestral population as a reference is not valid for models without separation of time scale, such as isolation by distance.

Ochoa & Storey [9] want to use the ancestral population as a reference, since their goal is to "consistently estimate IBD probabilities" despite their acknowledgement that "IBD probabilities are not absolute" (page 3, third paragraph). But, in addition to not being valid for models without separation of time scale [4], their definition requires the very strong assumption that alleles from the least related pair of individuals or populations in a study have zero IBD. If not, they estimate IBD relative to IBD in the least related pair of individuals or populations.

It seems better to us to acknowledge that IBD probabilities for target sets of alleles relative to IBD probabilities in a reference set of alleles are actually the parameters of interest and have estimators that are unbiased when large numbers of SNPs are used.

## A general population model

Following Nagylaki [32], we first present transition equations for mean coancestries between individuals, within and between populations, in a metapopulation where populations can differ in effective size and exchange migrants according to a specified migration matrix. This general model includes as special cases (i) the continent-islands model, where islands receive alleles / immigrants from an infinite sized continent; (ii) the finite islands model, where all islands exchange migrants with all others at the same rate, as well as (iii) the stepping stone model, where populations exchange migrants only with their neighbors(see Fig 1).

We assume a model where $r$ populations $i$, $i = 1, 2, \ldots, r$ of different effective sizes $N_i$ are interconnected by migration, with a general migration matrix $\mathbf{M}$. $N_i$ are the effective rather than the census sizes, hence allowing approximately for any reproductive system. Elements of the matrix $m_{ii'}$ are the proportions of alleles in row population $i$ that are from column population $i'$ in the previous generation, including the case of $i = i'$. The only constraint on $\mathbf{M}$ is that all its elements are positive or zero, and each row sums to 1. The coancestries for distinct pairs of individuals in each population and between pairs of populations at time $(t + 1)$ depend on their values in the previous generation (at time $t$) according to:

$$\theta_{ii'}^{(t+1)} = (1 - \mu)^2 \left( \sum_k m_{ik} m_{i'k} \phi_k^t + \sum_k \sum_{l \neq k} m_{ik} m_{i'l} \theta_{kl}^t \right) \tag{2}$$

where $\theta_{ii'}^t$ is the mean IBD probability for distinct pairs of individuals, one in population $i$, one in population $i'$, at time $t$, $\mu$ is the mutation rate, $\phi_i = 1/(2N_i) + (2N_i - 1)\theta_i/(2N_i)$ (we write $\theta_i$ or $\theta_{ii}$ for the mean IBD probability for distinct pairs of individuals both in population $i$) and $m_{ii} = 1 - \sum_{i' \neq i} m_{ii'}$. The first term in Eq 2 describes how the coancestry between populations $i$ and $i'$ at generation $t + 1$ depends on the proportion of immigrants received from population $k$

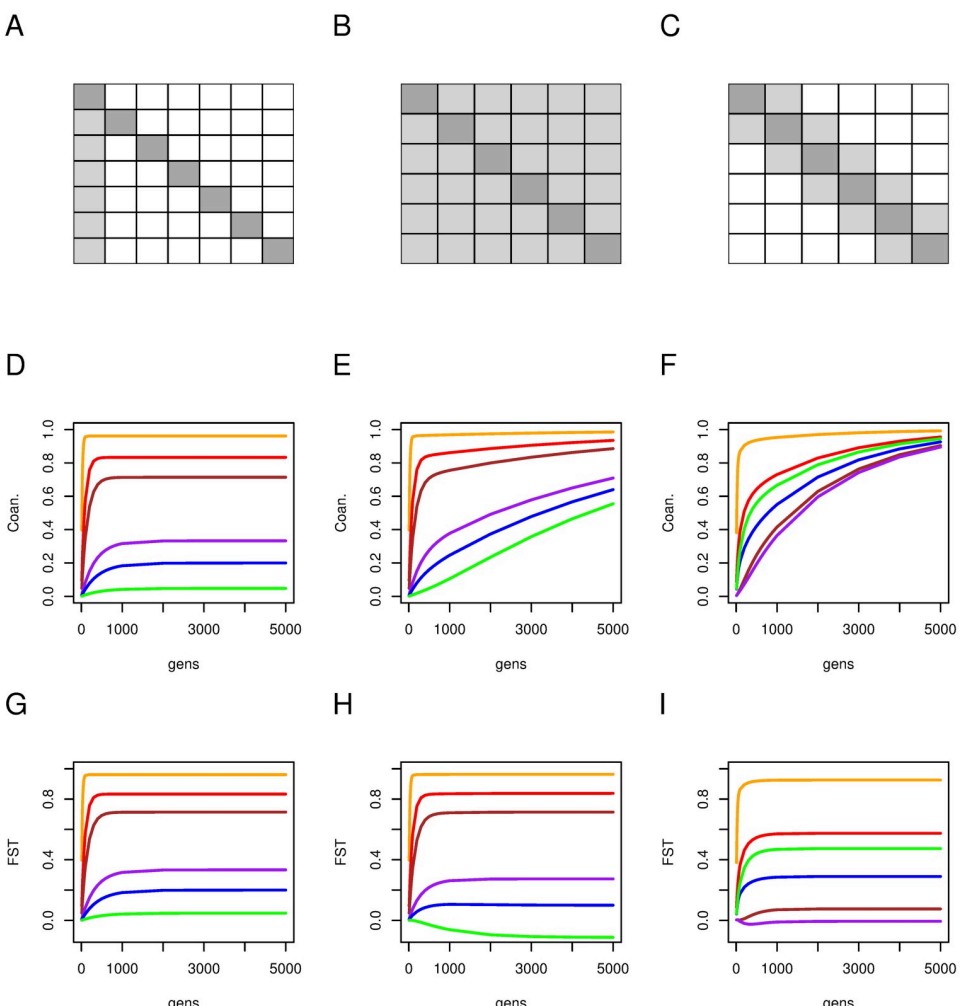

**Fig 1. Migration, coancestries and $F_{ST}$ for three migration models.** The mutation rate for all models is set to $\mu = 10^{-8}$. The top row shows three migration matrices for sets of six populations, a continent-islands (panel A), where the continent (leftmost column, $N = 10^9$) sends migrants to all other columns, and receives none, and six islands of sizes $N = 10, 50, 100, 500, 1000, 5000$ (orange, red, brown, purple, blue, green respectively) only receive immigrants from the continent at rate $m = 0.001$; a finite island (panel B) where each island ($N = 10, 50, 100, 500, 1000, 5000$; orange, red, brown, purple, blue, green respectively) sends and receives the same proportion $m = 0.001/5$ to/from all other islands; and a finite one dimensional (1D) stepping stone model(panel C) where populations of size $N = 10, 50, 1000, 1000, 100, 100$ (orange, red, brown, purple, blue, green respectively) receive a proportion $m = 0.01/2$ of immigrants from their left and right neighbours. White: cells with zero migration; light grey: cells with a positive migration term; dark grey: self. Middle row: Dynamics of within-population coancestries for The continent islands model (D), the finite islands model (E) and the stepping-stone model (F) through time, for the six different populations. Bottom row: Dynamics of population-specific $F_{ST}$ for the continent islands model (G), the finite islands model (H) and the stepping-stone model (I) through time, for the six different populations.

and the population size and coancestry of population $k$ at generation $t$, while the second term accounts for changes in coancestry between $i$ and $i'$ at generation $t + 1$ due to immigration from population $k$ and $l \neq k$ and their coancestry at generation $t$.

Because $N_i$ in the expression for $\phi_i$ is the effective size, where by definition mating occurs at random, any sex ratio bias or departure from random mating is accounted for, and since $\theta_{ii}$'s are for pairs of alleles from different individuals, we need not worry about self coancestry. The model is valid for gametic migration and also for zygotic migration providing sampling occurs

after dispersal. These equations can be rewritten in matrix form as:

$$\boldsymbol{\Theta}^{t+1} = (1-\mu)^2 \left[ \mathbf{M} \left( \mathbf{I}(\frac{1}{2}\mathbf{N}^{-1}) + \left[ \mathbf{J} - \mathbf{I}(\frac{1}{2}\mathbf{N}^{-1}) \right] \circ \boldsymbol{\Theta}^t \right) \mathbf{M}^{\mathbf{T}} \right] \tag{3}$$

where $\mathbf{J}$ is a matrix of 1's, $\mathbf{I}$ is the identity matrix, $\mathbf{M}^{\mathbf{T}}$ is the transpose of $\mathbf{M}$, $\mathbf{N}^{-1}$ is a vector with elements $1/N_i$ and $\circ$ denotes the Hadamard, or term by term, product of two matrices.

$\boldsymbol{\Theta}$ in any generation contains the coancestries relative to those at time $t = 0$. To obtain $\mathbf{F}_{ST}$, the matrix of coancestries *relative to* the mean coancestry for pairs of alleles, one in each of two different populations, we take the average, $\theta_B$, of the off-diagonal elements of $\boldsymbol{\Theta}$, subtract it from $\boldsymbol{\Theta}$, and divide the result by $1 - \theta_B$:

$$\boldsymbol{F_{ST}}^t = \frac{\boldsymbol{\Theta}^t - \mathbf{J}\theta_B^t}{1 - \theta_B^t} \tag{4}$$

which has the same form as Eq 1. Averaging the diagonal elements of $\mathbf{F}_{ST}$ over populations in any generation provides the overall $F_{ST} = (\theta_S - \theta_B)/(1 - \theta_B)$, and $\theta_S$, $\theta_B$ are the average coancestries for a random pair of individuals within populations and between population pairs, respectively, for that generation. This was also given by [5, 14, 17, 18, 23]. The off-diagonal elements of $\mathbf{F}_{ST}$ have an average zero by construction.

Eq 4 shows that $F_{ST}$ for a pair of populations $i$, $i'$ involves all pairs of population-pair coancestries because of the $\theta_B$ term. The commonly used pairwise $F_{ST}$, however, considers only a particular pair of populations $i$, $i'$ and the quantity then being addressed is $[(\theta_i + \theta_{i'})/2 - \theta_{ii'}]/(1 - \theta_{ii'})$.

Classical population genetic models can be derived from this general one as follows:

- The continent-island model is obtained by setting the size of, e.g. the first population, to $\infty$, and allowing for migration terms only on the first column of the migration matrix (migration from the continent to the islands). The diagonal of $\mathbf{M}$ will thus consist of a 1 for the first element, and $(1 - m_{i1})$ for the other diagonal elements, where $m_{i1}$ is the proportion of immigrants in island $i$ from the continent in the previous generation. For this model, Eq 2 can be rewritten:

$$\theta_i^t = (1-\mu)^2 (1-m_i)^2 \left[ \frac{1}{2N_i} + (1 - \frac{1}{2N_i})\theta_i^{(t-1)} \right]$$

At equilibrium between mutation migration and drift, this gives (e.g. [3]):

$$\theta_i = \frac{(1-m_i)^2(1-\mu)^2}{2N_i - (2N_i-1)(1-m_i)^2(1-\mu)^2} \approx \frac{1}{1 + 4N_i(m_i + \mu)}$$

- The finite island model is obtained by setting all off-diagonal terms of the migration matrix to $m/(r-1)$, where $m$ is the immigration rate and $r$ the number of populations, and the diagonal elements to $1 - m$.

- A 1-dimensional stepping stone model is obtained by setting the migration rate to 0 everywhere but to the left and right neighbour, whose rates are each set to $m/2$, and on the diagonal, whose value is set to $(1 - m)$. Populations at the left (or right) end of the set receive alleles only from the right (or left) at rate $m/2$.

Analytical solutions for the dynamic of coancestries in the finite island model and the stepping stone model can be found in [31] and [33] respectively.

We illustrate with these three models how $\boldsymbol{\Theta}$ and $\mathbf{F}_{ST}$ change through time for sets of six populations. Fig 1 shows the migration matrix (top row) through time for the continent island (panel A), the finite island model (panel B) and the stepping stone model (panel C). The populations all differ in effective sizes, varying from 10 to 5,000. The middle row (panels D, E, F) shows the diagonal elements of coancestries $\boldsymbol{\Theta}^t$ (the within-population coancestries) while the bottom row (panels G, H, I) shows the diagonal elements of $\mathbf{F}_{ST}^t$ (the population-specific $F_{ST}$'s) through time.

For the continent-islands model (left column), $\boldsymbol{\Theta}$ (panel D) reach equilibrium before 1000 generations, and off-diagonal elements are almost 0 (see data provided from GitHub site), thus $\mathbf{F}_{ST}$ shows the same dynamic as mean coancestries. Small populations have larger mean coancestries and $F_{ST}^i$ than large ones.

For the finite island model (middle column), we see a different pattern for $\boldsymbol{\Theta}$, as diagonal elements for the larger populations do not reach equilibrium within 5,000 generations (panel E), and the between-population mean coancestries are also positive (see data provided from GitHub site). On the other hand, all elements of $\mathbf{F}_{ST}$ have reached equilibrium (panel H), and we note that $F_{ST}$ for the largest population is negative, implying a random pair of individuals from this population share fewer alleles on average than a random pair with one individual from one population and the second from another. The off-diagonal elements of $\mathbf{F}_{ST}$ also differ from 0, with elements between the largest populations being negative and those between small populations being positive (see data provided from GitHub site).

For the 1-D stepping stone model (right column of Fig 1) we also see elements of the diagonal of $\boldsymbol{\Theta}$ increasing through time (panel F), as do elements of the off-diagonal elements (see data provided from GitHub site), although at a slower pace. On the other hand, all elements of $\mathbf{F}_{ST}$ (panel I) have reached an equilibrium value.

## A method of moments estimator for $F_{ST}$

We now describe the allele-sharing moment estimator of individual inbreeding, kinship and $\mathbf{F}_{ST}$ we first proposed in [18]. We allow for inbreeding (or any other form of non-random mating, such as clonal reproduction) within populations. We index populations with superscripts $i, i', \ldots$ and individuals with subscripts $j, j', \ldots$. The estimators are obtained from an $n_T \times n_T$ matrix $\mathbf{A}$ ($n_T = \sum_i^r n_i$, where $n_i$ is the sample size for population $i$ and $r$ the number of sampled populations) of allele-sharing among individuals. For a bi-allelic diploid locus (see S2 Text for a generalization to any ploidy level), allele-sharing between two individuals is 1 if the two individuals are homozygous for the same allele type, 0 if they are homozygous for different types, and 0.5 if at least one individual is heterozygous. Self-sharing is 1 for homozygous individuals and 0.5 for heterozygotes. When averaged over a large number of SNPs, this gives $A_{jj'}^{ii'}$, the allele-sharing between individuals $j$ and $j'$ in populations $i$ and $i'$ respectively. Populations $i, i'$ and individuals $j, j'$ may or may not be the same. A moment estimator of self and between pairs of individuals kinship [34] is obtained as

$$\hat{\mathbf{K}}_{AS} = \frac{\mathbf{A} - \mathbf{J}A_B}{1 - A_B} \tag{5}$$

where $A_B$ is the average of all the off-diagonal elements of $\mathbf{A}$ and $\mathbf{J}$ is an $n_T \times n_T$ matrix of 1s. Note the form of Eq 5 is identical to Eqs 1 and 4. Individual inbreeding coefficients relative to the total population are then estimated as $\hat{F}_{AS_j} = 2\hat{k}_{AS_{jj}} - 1$ [35].

The mean allele-sharing statistics between individuals within ($A^{ii}$) and between ($A^{ii'}$) populations are obtained by averaging individual $A_{jj'}^{ii'}, j \neq j'$ (thus self allele-sharing is excluded from

these averages) for each population $i$ and population pair $i$, $i'$. These $(A^{ii'})$ are stored in an $r \times r$ matrix $\bar{\mathbf{A}}$.

The individual inbreeding coefficient of individual $j$ in population $i$, $\hat{F}^i_j$, relative to the mean coancestry of population $i$ is obtained as:

$$\hat{F}^i_j = 2\left(\frac{A^{ii}_{jj} - A^{ii}}{1 - A^{ii}}\right) - 1 \tag{6}$$

The average of the $\hat{F}^i_j$s from one population gives an estimator of population $i$ mean inbreeding coefficient $\hat{F}^i_{IS}$. Averaging in turn these $\hat{F}^i_{IS}$ over populations lead to the overall estimator of within-population inbreeding coefficient $\hat{F}_{IS}$, which, for equal sample sizes, is identical to Weir & Cockerham estimator of $F_{IS}$ [36].

Writing as $\bar{A}_B$ and $\bar{A}_S$ the averages of the off-diagonal and diagonal elements of $\bar{\mathbf{A}}$, respectively, population-specific $F_{ST}$ estimates are obtained as:

$$\hat{F}^i_{ST} = \frac{A^{ii} - \bar{A}_B}{1 - \bar{A}_B} \tag{7}$$

The average over populations of these $\hat{F}^i_{ST}$ is an overall estimator of $F_{ST}$:

$$\hat{F}_{ST} = \frac{\bar{A}_S - \bar{A}_B}{1 - \bar{A}_B} = \frac{1}{r}\sum_i^r \hat{F}^i_{ST}, \tag{8}$$

which, if all sample sizes are equal, is identical to Weir & Cockerham $F_{ST}$ estimator for genotypes [36].

From matrix $\bar{\mathbf{A}}$, we can also obtain the following quantities:

$$\hat{F}^{ii'}_{ST} = \frac{A^{ii'} - \bar{A}_B}{1 - \bar{A}_B}, \tag{9}$$

the mean coancestry for individuals, one in population $i$ and one in population $i'$, relative to the average mean coancestry between pairs of populations. We are not aware of studies making use of the allele-sharing quantity described by Eq 9 although, for equal sample sizes, numerical values will be the same as those described by Weir & Hill [14]. Mary-Huard & Balding [19] described a similar quantity for tree-like population structure, but did not explore its properties in detail.

For pairs of populations $i$, $i'$, using only these two populations

$$\hat{F}^{ii'}_{STp} = \frac{(A^{ii} + A^{i'i'})/2 - A^{ii'}}{1 - A^{ii'}}, \tag{10}$$

gives the average mean coancestry within populations $i$ and $i'$ relative to the mean coancestry between populations $i$ and $i'$. Eq 10 gives the expression for pairwise $F_{ST}$ often used to assess, for instance, whether isolation by distance is occurring in a dataset [20, 33].

$\hat{F}^i_{ST}$ and $\hat{F}^{ii'}_{ST}$ are conveniently stored in an $r \times r$ matrix $\hat{\mathbf{F}}_{ST}$:

$$\hat{\mathbf{F}}_{ST} = \frac{\bar{\mathbf{A}} - \mathbf{J}\bar{A}_B}{1 - \bar{A}_B} \tag{11}$$

$\hat{\mathbf{F}}_{STp}$, the matrix of pairwise $F_{ST}$ values ($F_{STp}$, Eq 10), can be retrieved instead from $\hat{\mathbf{F}}_{ST}$ by replacing the $A^{ii'}$s in Eq 10 with the corresponding $F^{ii'}_{ST}$ elements of $\hat{\mathbf{F}}_{ST}$.

While mean within and between population allele-sharing $\bar{\mathbf{A}}$ is not an unbiased estimator of mean within and between population coancestries $\mathbf{\Theta}$ and $\bar{A}_B$ is not an unbiased estimator of $\theta_B$, $\hat{\mathbf{F}}_{ST}$ (Eq 11) is an unbiased estimator of $\mathbf{F}_{ST}$ (Eq 4) [18] as we will show empirically.

**Another moment estimator of $F_{ST}$.** Ochoa & Storey [9] recently proposed a closely related quantity. Rather than measuring average between-individual allele-sharing within populations relative to average allele-sharing between pairs of populations, they use the minimum average allele-sharing between populations as a reference, suggesting this should estimate 'absolute mean coancestry relative to the most recent common ancestral population' and claiming the measured quantities estimate 'probabilities of Identity by Descent'. This new reference point changes the definition of $F_{ST}$ in Eq 4 to

$$\mathbf{F}_{ST}^{OS} = \frac{\mathbf{\Theta} - \mathbf{J} \min(\theta^{i \neq i'})}{1 - \min(\theta^{i \neq i'})} \tag{12}$$

and we note that moving from one definition to the other is straightforward:

$$\mathbf{F}_{ST}^{OS} = \frac{\mathbf{F}_{ST} - \mathbf{J} \min(F_{ST}^{i \neq i'})}{1 - \min(F_{ST}^{i \neq i'})}; \quad \mathbf{F}_{ST} = \frac{\mathbf{F}_{ST}^{OS} - \mathbf{J} \bar{F}_{ST}^{i \neq i' OS}}{1 - \bar{F}_{ST}^{i \neq i' OS}}$$

and can be extended to any constant one might want to use as a reference point (see S1 Text). The pairwise quantities estimated with Eq 10 do not change with reference point.

## Statistical properties of $\hat{\mathbf{F}}_{ST}$

In order to investigate the statistical properties of $\mathbf{F}_{ST}$ estimators, we first simulated genetic data sets with different population structures. We simulated genotype frequencies at 10, 000 independent loci from a series of populations connected by migration. Populations can have different sizes and migration rates between populations are entered in matrix form, allowing among other things the simulation of the classical models of population structure described previously. For these simulations, we used the function `sim.genot.metapop.t` from the R package `hierfstat 0.5-11` [37, 38].

We focus on $F_{ST}$ estimators for which the underlying model includes possible non-independence of populations. This excludes estimators based on the strict $F$ model [25, 26], as that model assumes all off-diagonal elements of $\mathbf{F}_{ST}$ to be zero. On the other hand, estimators based on the admixture $F$ model (AFM, [17]) would be relevant.

The function `fs.dosage` from `hierfstat` was used to obtain the moment estimator $\hat{\mathbf{F}}_{ST}$. We used the RAFM package [39] and its `do.all` function with a burn-in of 5, 000 steps and a chain length of 10, 000, saving every fifth step to obtain the Bayesian AFM coancestry estimates. The last 200 saved steps from the MCMC were averaged to obtain the Bayesian estimates of $\mathbf{F}_{ST}$ using Eqs 7 and 9. We checked in all cases the chain trace to insure it had converged. Predicted values for $\mathbf{F}_{ST}$ are obtained using Eqs 3 and 4.

We use Root Mean Square Error (RMSE), the square root of the mean of squared deviations of the estimators from their expected values, to evaluate the different estimators and sampling strategies.

For simulations run with `sim.genot.metapop.t`, we also estimated $\mathbf{F}_{ST}$ using the moment estimator with subsets of 1, 000 and 100 loci (100 replicates for each). For the Bayesian estimator, we used only 100 loci, as running time for more loci was prohibitive.

Nowadays most data sets consist of several thousand SNPs distributed throughout the genome of the studied organisms and the genetic map length in a given species then becomes the ultimate constraint in the number of independent markers we can examine. In order to

investigate the effects the genetic map length, the number of loci on the map and the number of individuals used, have on the bias and variance of $\hat{\mathbf{F}}_{ST}$, we ran a second set of simulations using a coalescent-with-recombination algorithm with the program msprime [40].

RMSE for data simulated with msprime were compared to the RMSE of data simulated with sim.genot.metapop.t with 10, 000 loci.

Finally, we estimated RMSE from the 1, 000 Genome data set [28]. We used as expectations for this data set the values obtained from the 77, 818, 345 SNPs (no filtering) and the 2, 504 individuals. The effect of subsampling loci was tested by subsampling uniformly $10^5$, $10^4$, $10^3$, $5 \times 10^2$ and $10^2$ SNPs per chromosome keeping all populations and individuals, while the effect of subsampling individuals was tested using 50, 20, 10, 5 or 2 individuals per population and keeping all loci.

## Simulated population structures

We simulated three scenarios with different population structures. For each scenario, we simulated 20 replicates with 10, 000 loci, sampling 50 individuals using sim.genot.metapop.t.

- A finite island model with 10 islands with different sizes: $N_1 = N_2 = 1000$; $N_3 = N_4 = 10$; $N_5 = N_6 = 100$; $N_7 = N_8 = 500$; $N_9 = N_{10} = 2000$, and a migration rate $m = 0.001$.

- A one-dimensional stepping stone model with 10 populations, constant population sizes $N = 1000$ and $m = 0.02$ between adjacent populations.

- A river system with 2 tributaries rivers (see Fig 2 for details).

We ran a second set of simulations for this river system structure with the program msprime [40]. For this second set of simulations, we generated five replicates of a genome made of 20 chromosomes each $10^8$ base pairs long and with a recombination rate between adjacent base pairs of $10^{-8}$, thus one crossing over per chromosome on average.

The Code to generate the simulated data set can be found at https://github.com/jgx65/PlosGenetPopulationFST.

## Verification and comparison

### Simulated data

**Results for the finite island model** made of 10 populations with different sizes are shown Fig 3. Panel A shows the expected $\mathbf{F}_{ST}$. The smaller populations (3 and 4) show the largest $F_{ST}^i = 0.96$ while the largest (9 and 10) have $F_{ST}^i = 0.05$. Off-diagonal elements of $\mathbf{F}_{ST}$ are all fairly small between −0.06 and 0.09, those between large populations are slightly negative while those between small populations are positive.

Panel B shows RMSE distribution of $\hat{\mathbf{F}}_{ST}$ according to the sampling scheme or estimator used. We see a large effect of having fewer loci. Panels C and E illustrate that by using $10^3$ rather than $10^4$ SNPs, the variance of the estimates increases (but estimators remain unbiased), while subsampling individuals has almost no effect on the variance of the estimate (illustrated with panel D where only two out of 50 individuals are sampled).

The Bayesian estimates (panel E) show large biases, with an underestimation of large values and an overestimation of low values of the elements of $\mathbf{F}_{ST}$. With 1, 000 loci, the variability around each point estimate is similar between the method of moments and the Bayesian method.

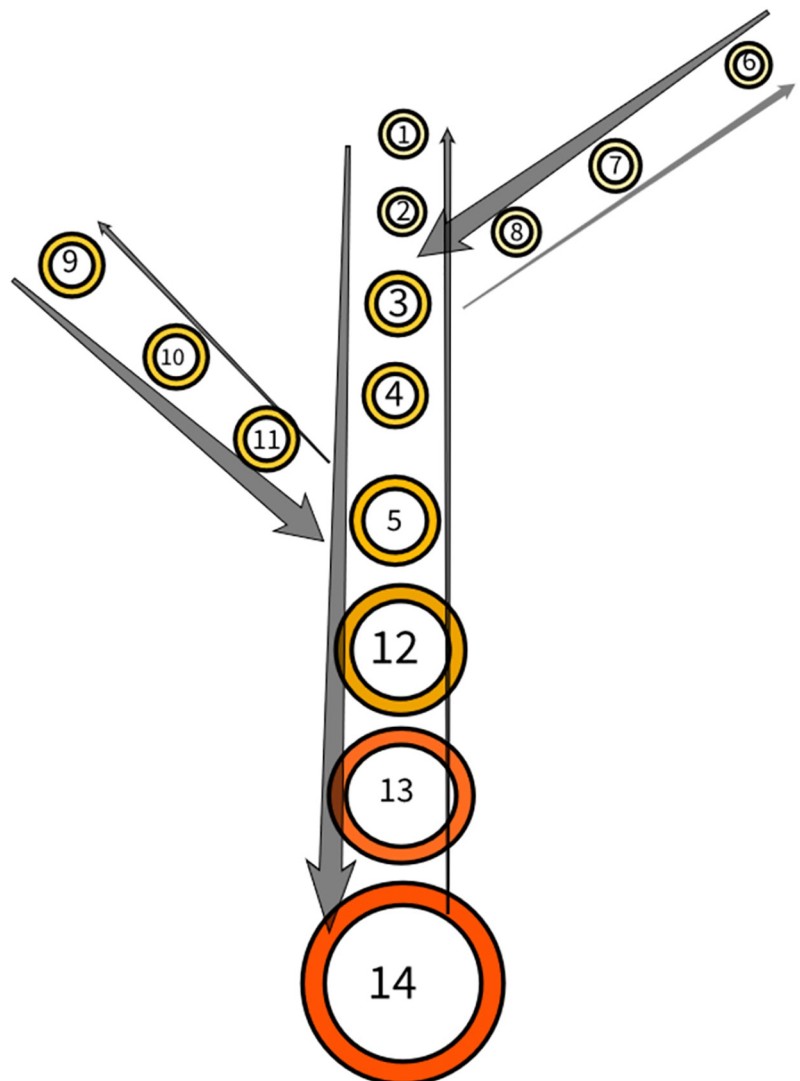

**Fig 2. Sketch for the river system simulation.** Each circle is a site, and size and colour indicate population size. The main river is made of stations 1 to 5 and 12 to 14. The first tributary river is made of stations 6 to 8 and joins the main river at station 3. The second tributary is made of stations 9 to 11 and joins the main river at station 5. Migration downstream to the nearest neighbour is $m_d = 0.02$ and migration upstream to the nearest neighbour is four times less, $m_u = 0.005$. Site 3 receives from and sends migrants to sites 2 and 8, and site 5 receives from and sends migrants to sites 4 and 11. Population sizes increase as stations get closer to the mouth of the river, with $N_{1-2,6-8} = 100$; $N_{3-4,9-11} = 200$; $N_5 = 400$; $N_{12} = 800$; $N_{13} = 1,000$; $N_{14} = 5,000$.

**Results for the one-dimensional stepping stone model are shown Fig 4.** Panel A shows the expected $\mathbf{F}_{ST}$. Populations at the end of the stepping stone show the largest $F_{ST}^i$ and those at the centre have positive but lower values. Off-diagonal elements of populations far apart are negative, the further apart the more negative.

Panel B shows RMSEs of $\hat{\mathbf{F}}_{ST}$ for a varying number of loci and individuals. We see the same pattern as was seen for the island model: the fewer loci, the larger are RMSE, but with no bias for the moment estimator (Panel C and E). Sub-sampling five individuals per population out of 50 has almost no effect on RMSE values (panel D).

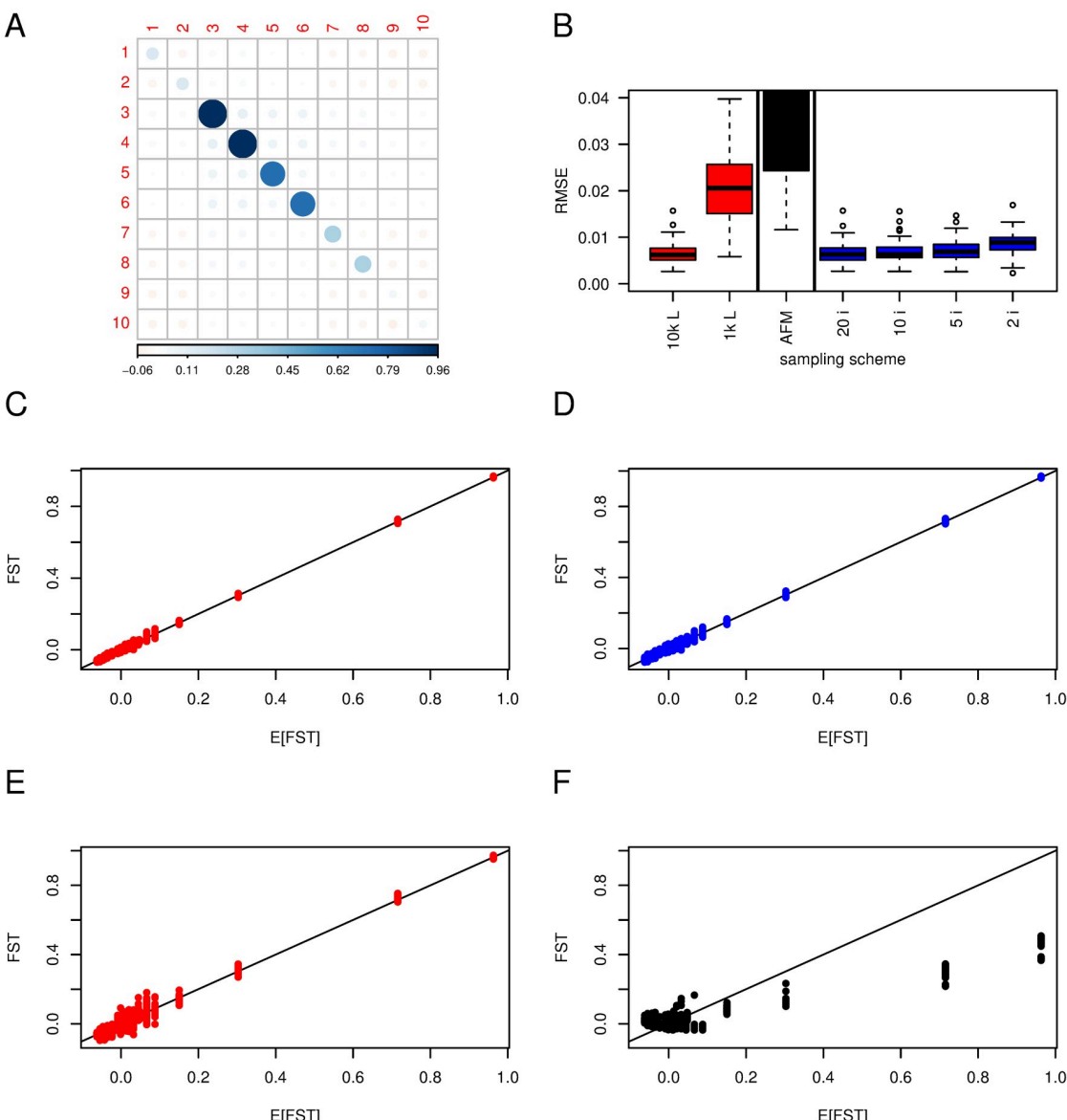

**Fig 3. $\hat{F}_{ST}$ in a finite island model.** 10 populations with different effective sizes: $N_1 = N_2 = 1,000$; $N_3 = N_4 = 10$; $N_5 = N_6 = 100$; $N_7 = N_8 = 500$; $N_9 = N_{10} = 2,000$, and a migration rate $m = 0.001$. The default sample size and number of loci for estimates are 50 individuals in each population and $10^4$ SNPs. Panel A shows the expected ($F_{ST}$) after 2,000 generations from the transition equations (Eq 4); The darker and the larger the circle, the larger the elements, either positive (blue) or negative (red); scale at the bottom. Panel B shows the distributions of Root Mean Square errors (RMSE) for all the elements of $\hat{F}_{ST}$ based on 20 or 100 replicates. '10k L, 1k L': subsampling of $10^4$, $10^3$ SNPs respectively (red); 'AFM': Bayesian estimator from AFM, based on 1,000 SNPs (black); '20 i, 10 i, 5 i, 2 i': Subsampling of 20, 10, 5 or 2 individuals, $10^4$ SNPs (blue); the vertical bars separate the different sampling schemes / estimates. The four lower panels show the relation between expected and estimated $F_{ST}$ for $10^4$ SNPs (panel C), $10^4$ SNPs, 2 individuals (panel D), $10^3$ SNPs (panel E), AFM method with $10^3$ SNPs (panel F). Red color illustrates subsampling of loci, blue subsampling of individuals and black the Bayesian estimator.

The Bayesian estimates based on 100 loci (panel F), while less variable for each point estimate than the method of moments (compare panel E and F), are strongly biased, with overestimation for low values of the parameter and underestimation for high values, and most estimates staying close to 0.

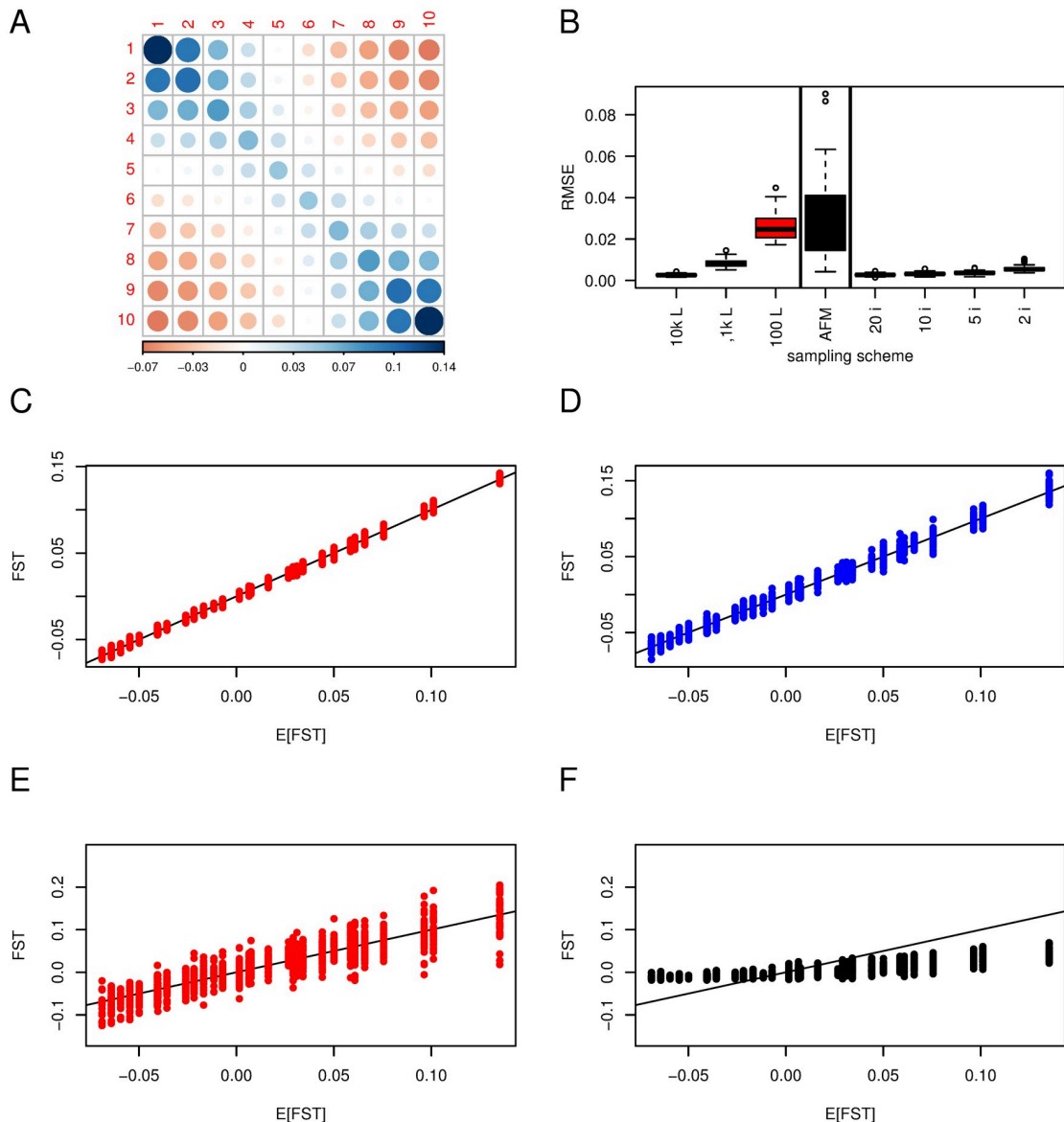

**Fig 4. $\hat{\mathbf{F}}_{\mathbf{ST}}$ in a 1D stepping stone model.** 10 populations with $N = 1,000$; $m = 0.005$ between adjacent populations. The default sample size and number of loci for estimates are 50 individuals in each population and $10^4$ SNPs. Panel A shows the expected $\mathbf{F}_{\mathbf{ST}}$ after 4,000 generations from the transition equations (Eq 4); The darker and the larger the circle, the larger the elements, either positive (blue) or negative (red); scale at the bottom. Panel B shows the distributions of Root Mean Square errors (RMSE) for all the elements of $\hat{\mathbf{F}}_{\mathbf{ST}}$ based on 20 or 100 replicates. '10k L, 1k L, 100 L': $10^4$, $10^3$ and 100 SNPs respectively (red); 'AFM': 100 loci, using the Bayesian estimate from AFM (black); '20 i, 10 i, 5 i, 2 i': Subsampling of 20, 10, 5 or 2 individuals, $10^4$ loci (blue); the vertical bars separate the different sampling schemes / estimates. The four lower panels show the relation between expected ($E[F_{ST}]$) and estimated $F_{ST}$ for $10^4$ SNPs (panel C); $10^4$ SNPs but only two individuals (panel D); 100 SNPs (panel E); and 100 SNPs, AFM method (panel F); red color shows the effect of subsampling of loci, blue subsampling of individuals and black the Bayesian estimates.

**Results for the river system where populations differ in size and migration is asymmetric, are shown Fig 5.** Panel A shows the expected $\mathbf{F}_{\mathbf{ST}}$. The largest $F_{ST}^i$ are for stations 1 and 6, those close to the sources of the river system, while the smallest is for population 14, at the mouth of the river. The off-diagonal elements of the tributaries further up the river are larger

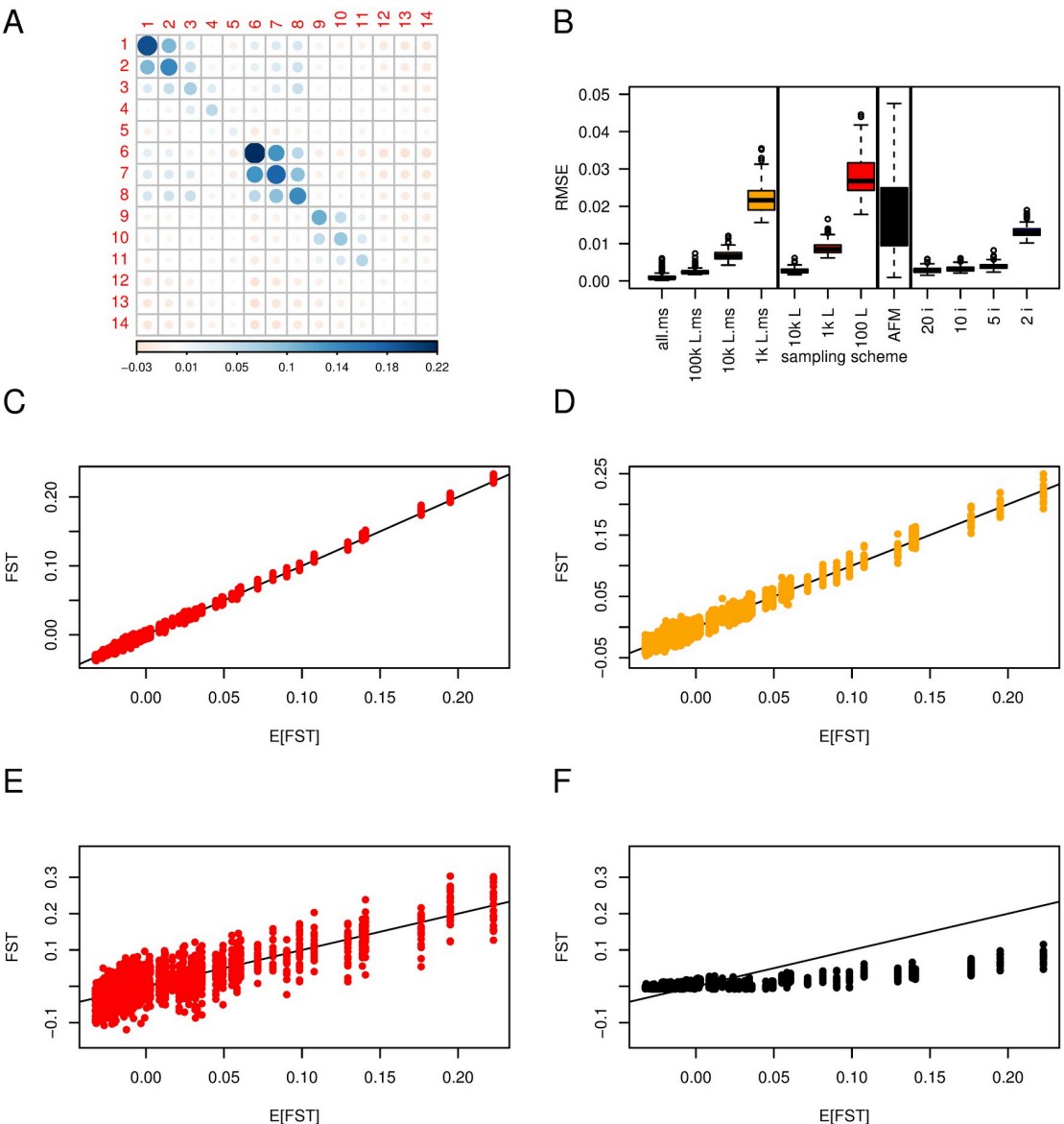

**Fig 5. $\hat{F}_{ST}$ in a river system.** See methods for a description of the system. The default sample size and number of loci for estimates are 50 individuals in each population and $10^4$ SNPs. Panel A shows the expected $\mathbf{F}_{ST}$ after 2, 000 generations from the transition equations (Eq 4); The darker and the larger the circle, the larger the elements, either positive (blue) or negative (red); scale at the bottom. Panel B shows the distributions of Root Mean Square errors (RMSE) for all the elements of $\hat{\mathbf{F}}_{ST}$ based on 20 or 100 replicates. 'all.ms, 100kLms, 10kLms, 1kLms': All, $10^5$, $10^4$, $10^3$ SNPs from `msprime` simulations (yellow); '10k L, 1k L, 100 L': $10^4$, $10^3$ and 100 SNPs respectively (red); 'AFM': 100 loci, using the Bayesiann estimator from `AFM` (black); '20 i, 10 i, 5 i, 2 i': Subsampling of 20, 10, 5 or 2 individuals, $10^4$ loci (blue); the vertical bars separate the different sampling schemes / estimates. The four lower panels show the relation between expected and estimated $F_{ST}$ for $10^4$ SNPs (panel C), $10^4$ SNPs from `msprime` simulations (panel D), 100 SNPs (panel E), 100 SNPs, AFM method (panel F).

than other elements, and negative elements appear for pairs where one member is close to the mouth and the other is in one of the tributaries.

The effect of subsampling independent loci is similar to what we have seen for the island and stepping stone model, with RMSEs increasing with fewer loci (red boxplots, panel B, and

Panel C and E panels for $10^4$ and 100 SNPs respectively). Subsampling individuals has little effect on RMSEs but for two individuals per population, where RMSEs increase slightly.

The Bayesian estimator again gives consistently biased estimates, with an underestimation of large and an overestimation of small elements (panel F). The lower RMSEs for the Bayesian estimates seen on panel B, (black boxplot) is due to many elements of $\mathbf{F}_{ST}$ being close to 0, and, since the Bayesian estimates are less variable than the moment estimates with 100 loci but are biased toward 0, they show a smaller RMSEs.

Finally for this population structure, we also simulated data with `msprime` to look at the effect of linkage among SNPs (orange boxplot in panel B). Compared with the simulations with independent SNPs, roughly 10 times more SNPs are necessary to obtain similar RMSEs. Panel C and D show the precision of the estimates for $10^4$ SNPs, with independent SNPs (panel C) and partially linked SNPs (panel D), where the variation is larger than with independent SNPs.

One important way in which the allele-sharing estimator differs from those of Weir & Cockerham (WC) [36] and Weir & Hill [14] is in assigning equal weights to all samples, even if their sizes differ (if samples sizes are equal, the mean of the diagonal elements of $\hat{\mathbf{F}}_{ST}$ is identical to WC's $\hat{F}_{ST}$). In S3 Text, we discuss and illustrate the effects unequal sample sizes and only a portion of the populations sampled have on $\hat{\mathbf{F}}_{ST}$. S1 Fig shows even very heterogeneous sample sizes provide unbiased and low variance estimates of $\mathbf{F}_{ST}$. Sampling in only half the populations also leads to unbiased and low variance estimates of $\mathbf{F}_{ST}$.

## Applications

### Estimates of $\mathbf{F}_{ST}$ in the 1, 000 genomes

Results from the 1, 000 genomes project [28] with estimates of $\mathbf{F}_{ST}$ and pairwise $\mathbf{F}_{ST_p}$ are shown Fig 6. Looking at panel A ($\hat{\mathbf{F}}_{ST}$) we see all estimates where at least one of the populations is from Africa are negative, while all other estimates are positive. A negative value of $\hat{F}_{ST}^{ii'}$ implies that the pairs of populations considered have less allele-sharing than random pairs from the whole world. As African genomes are more heterozygous, it is not surprising they show the lowest allele-sharing with other African (including those from their own population) or non-African genomes. Populations from East Asia shows the largest $\hat{F}_{ST}^{i}$ values, followed by European and South Asians. Admixed American populations are the more heterogeneous, with Puerto Ricans from Puerto Rico (PUR) showing the lowest values and Peruvian from Lima (PEL) the highest (S2 Fig).

Panel B of Fig 6 shows pairwise $\hat{\mathbf{F}}_{ST_p}$. Here all values are positive (a property of $\hat{\mathbf{F}}_{ST_p}$), and all values for pairs of populations from the same continent are close to 0. African populations show the largest $\hat{F}_{ST_p}$ with populations from other continents, in particular East Asia and PEL. Among non-African populations, the largest differences are between East Asian and European samples.

Panel C shows RMSE for subsampling loci in blue and individuals in red. Using fewer loci and individuals increases RMSE for $\hat{\mathbf{F}}_{ST}$.

The effect of subsampling individuals on RMSE for $\hat{\mathbf{F}}_{ST}$ differs from what we have seen in the simulations, as we observed some elements of the RMSE for $\hat{\mathbf{F}}_{ST}$ have larger values (S3 Fig). From panel D of Fig 6, we see as few as 10 individuals could give results almost identical to 100 in homogeneous populations, but this is not so for admixed populations, where we see a large variation among replicates when we subsample 10 individuals. This is because in admixed

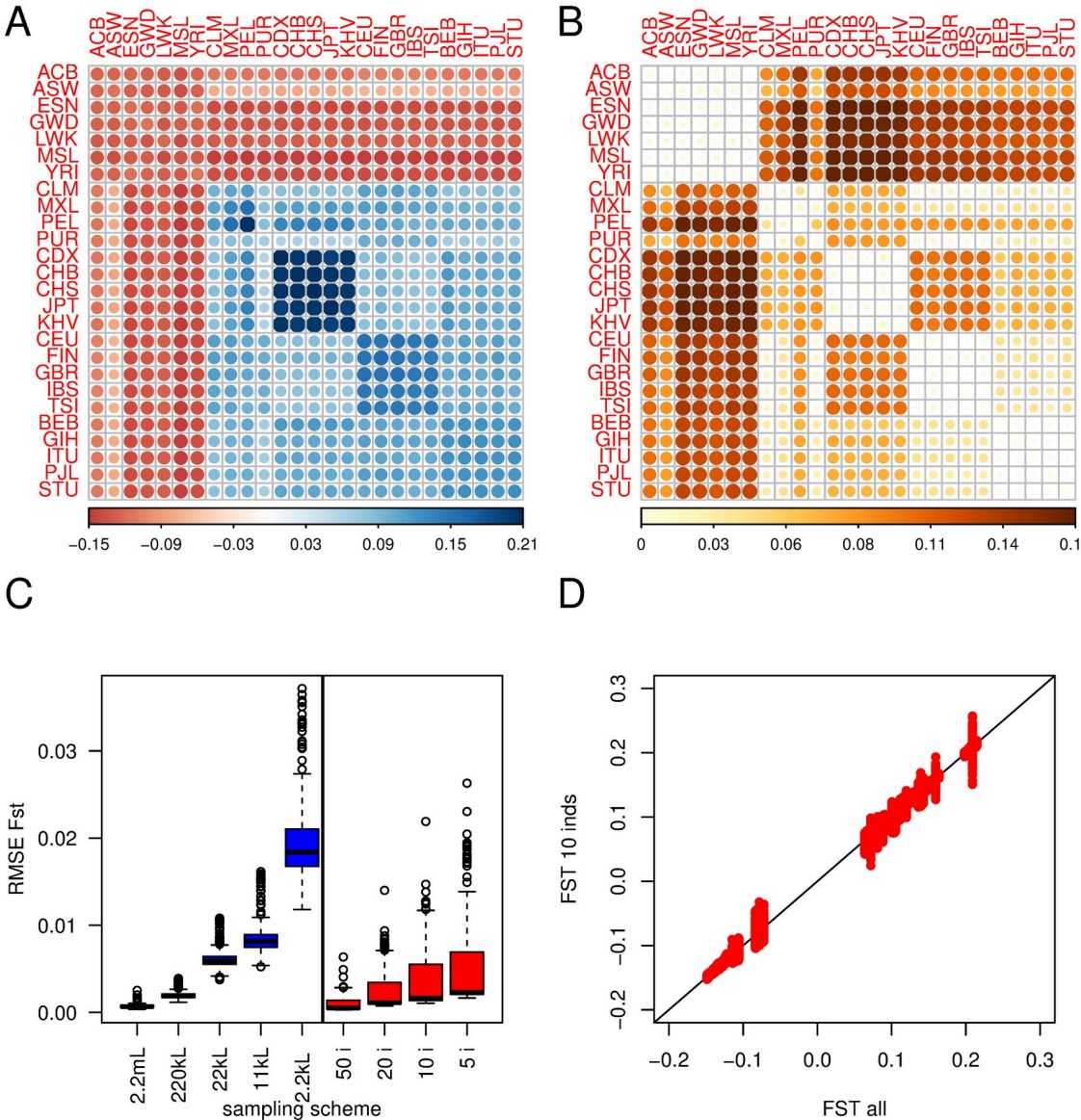

**Fig 6. Estimates of $\mathbf{F}_{ST}$ from the 1, 000 genomes.** The top row shows estimates based on the 2,504 individuals and 77, 818, 345 SNPs from phase 3 of the 1, 000 genomes project. Panel A shows $\hat{\mathbf{F}}_{ST}$; the darker and the larger the circle, the larger the elements, either positive (blue) or negative (red); scale on the right. Panel B shows pairwise $\hat{\mathbf{F}}_{ST_p}$, all positives; The darker the colour and the larger the circle, the larger the element; scale at the bottom. Panel C shows the distribution of RMSEs for $\hat{\mathbf{F}}_{ST}$. '2.2m L, 220k L, 22k L, 2.2k L' (in blue): subsampling of the corresponding number of SNPs from the total data set; '50 i, 20 i, 10 i, 5 i' (in red): subsampling of the corresponding number of individuals from the total dataset. Panel D shows $\hat{\mathbf{F}}_{ST}$ estimated from 10 individuals (100 replicates) against $\hat{\mathbf{F}}_{ST}$ estimated from all individuals.

populations the ancestry composition of the subsamples will vary and affect the corresponding elements of $\hat{\mathbf{F}}_{ST}$.

## Discussion

In this paper, we provide predicted values for $\mathbf{F}_{ST}$, the mean coancestries within and among populations relative to the average between populations mean coancestries, for a very general

model of population structure in a diploid species. We show our allele-sharing, moment-based estimator of $\mathbf{F}_{ST}$ to be efficient and unbiased in all simulation scenarios, and for all elements (on and off-diagonal) of the matrix. The Bayesian estimator [17] is biased for all simulated scenarios and does not scale to the size of genomic datasets generated nowadays (it took 24 hours to obtain the Bayesian estimates with 50 individuals per sampled population and 100 loci only, while it took seconds to obtain the corresponding moment estimates).

We find that as few as 10 or even five individuals per population are sufficient to estimate accurately $\mathbf{F}_{ST}$, unless the population contains admixed individuals. With $10^4$ SNPs, estimates are very accurate, and $10^3$ independent SNPs might actually be enough. However, one should be aware that if SNPs are linked, as may be the case with small genomes and/or small population sizes, more SNPs are needed, as we illustrate with the river-system example. An aspect of the sampling that was not explored here but deserves further scrutiny is low coverage sequencing, for instance using similar approaches to Hivert *et al.* [41].

In real data, it is unclear how linkage should be accounted for. In practice, either blocks of a constant number of SNPs or blocks of a constant number of nucleotides are used for bootstrapping, but it might be more appropriate to define block size according to the recombination rate of the different regions, and we note this is an area of active research [42]. Here, we obtain confidence intervals for continent and overall $\hat{F}_{ST}$ in the 1,000 genomes by bootstrapping estimates obtained from blocks of 100 kilobases, but recognized that this is *ad hoc* and an avenue for further research.

In deriving $\hat{\mathbf{F}}_{ST}$, we make no assumptions about mating system, inbreeding level, ploidy level or even whether reproduction is sexual [18]. As long as allele-sharing between individuals can be measured, $\hat{\mathbf{F}}_{ST}$ can be estimated.

## Use of $\hat{\mathbf{F}}_{ST}$

Global $F_{ST}$ and pairwise $\mathbf{F}_{ST_p}$ are commonly estimated in surveys of population structure, and population-specific $F_{ST}$'s have also been used [23, 26, 43]. But we are not aware of studies empirically estimating the off-diagonal elements of $\mathbf{F}_{ST}$.

Areas where such measures should be useful are molecular ecology and conservation genetics, as the off-diagonal elements inform about how much of the overall genetic diversity is captured by the pair of populations considered. Large positive values indicate that the two populations represent a small proportion of the overall diversity, while large negative values indicate to the contrary that these two populations together harbour as much as or more diversity than all populations together. For instance, in the stepping-stone example (Fig 4), populations 1 and 10 each contain less genetic diversity than others (they have large and positive $F_{ST}^{[1]}$ and $F_{ST}^{[10]}$, panel A), but are also those that together harbour the most diversity (large and negative $F_{ST}^{[1,10]}$, panel A). In the river system example (Fig 5), the populations harbouring the most diversity together (those with the lowest $F_{ST}$) would be $F_{ST}^{[1,14]}$ and $F_{ST}^{[6,14]}$, but we see that the pair 1 and 6, with the largest pairwise $F_{ST_p}$ (see data provided from GitHub site), does not have the lowest $F_{ST}$, and $F_{ST}^{[1,6]}$ is positive.

How large $F_{ST}^{[i,i']}$ is does not inform about how large $F_{ST_p}^{[i,i']}$ is: imagine two populations have fixed the same allele over the majority of the SNPs, but fixed different ones at a few loci. $F_{ST_p}^{[i,i']}$ would be one, but depending on the genetic make up of other populations, $F_{ST}^{[i,i']}$ could be very large, meaning these two populations together capture a small fraction of the overall diversity only. On the other hand, if the two populations have fixed alternate alleles at most of their loci,

$F_{ST_p}^{[i,i']}$ would still be one but $F_{ST}^{[i,i']}$ is likely going to be negative, as these two populations together would show maximum possible diversity.

## Choice of a reference point

Ochoa and Storey [9] suggest using the allele-sharing between the least related pair of populations, rather than the average between population allele-sharing, as a reference point. They claim this population pair with the lowest allele-sharing represents the closest to what would have been the ancestral population. By using the minimum between-population allele-sharing as a reference point, they ensure that all terms of $\mathbf{F}_{ST}^{OS}$ are positive because they want to interpret these values in terms of identity by descent. But we note their estimates are not for probabilities of identity by descent.

Karhunen & Ovaskainen [17] also estimate coancestries relative to an ancestral population, but the model they use, the admixture F model (AFM), is more restrictive than ours or Ochoa & Storey's [9], and they were careful in distinguishing between mean coancestries, estimated relative to the allele frequencies in the ancestral gene pool (see figure 1 and Eq 12 in [17]) and $F_{ST}$ (Eq 4 in [17]). While we see great values in obtaining estimates of mean coancestries relative to a reference population in the past (for instance in the context of detecting local adaptation on traits, see [44, 45]), we emphasise these coancestries differ from the standard definition of $F_{ST}$ given in Eq 1 [5, 17, 18, 46].

Using $\bar{A}_B$, the average of the off-diagonal elements of $\bar{\mathbf{A}}$, as a reference point allows for an easy interpretation of the individual $\hat{F}_{ST}^{[i,i']}, \forall i, i' \in [1, r]$: a negative value implies the corresponding pair is less related than a random pair from the total set, as we illustrated with the 1,000 genomes. Still (although we don't advocate it), using the method of moments, it would be straightforward to construct unbiased estimators of coancestries relative to any other reference point, as we show in S1 Text. For instance, one might use an external reference, which may be useful in a forensic context [21, 47], or when studying invasive species, where the population of origin of the invasive individuals might be a relevant reference point; or the median of the off-diagonal elements of $\bar{\mathbf{A}}$ instead of the mean, or some small percentile point instead of the minimum chosen by Ochoa & Storey [9] to avoid the undesirable statistical effects of using the minimum, as we now discuss.

While the algebraic difference in the formulae between our estimator and that proposed by Ochoa & Storey [9] is trivial, we show with data from the 1000 Genomes project (S3 Table and S4 Fig) that the two estimates can greatly differ (Overall $\hat{F}_{ST} = 0.083$ against overall $\hat{F}_{ST}^{OS} = 0.202$). While $\hat{F}_{ST}$ is similar to the previously reported $F_{ST}$ for human populations (e.g. [27]), values as large as $\hat{F}_{ST}^{OS}$ have not been reported. It is interesting to note that the textbook estimator of $F_{ST}$, $Var(p)/[\bar{p}(1 - \bar{p})]$, which, with these samples sizes (2.504 individuals, $\approx 100$ individuals per population and 26 populations), should be little affected by statistical biases, gives a value for the same dataset of 0.088 (range [0.083, 0.094]), much closer to $\hat{F}_{ST}$ than to $\hat{F}_{ST}^{OS}$.

Two other considerations, one theoretical and the other empirical, indicate that $\hat{F}_{ST}$ is to be preferred over $\hat{F}_{ST}^{OS}$:

Imagine a very large number of populations. All but two populations (1 and 2) contain only heterozygotes at all loci (we would obtain the same result with any genotypic composition maintaining an allelic frequency at 0.5 in all populations but the first two), and populations 1 and 2 are fixed for the opposite homozygotes at all loci. For populations 1 and 2, fixed for opposite homozygotes, $F_{ST}^{[1]}$ and $F_{ST}^{[1]OS}$ will be one, as will $F_{ST}^{[2]}$ and $F_{ST}^{[2]OS}$. $F_{ST}^{[1,2]}$ will tend to $-1$,

while $F_{\mathrm{ST}}^{[1,2]\mathrm{OS}}$ will be 0, by definition. All other $F_{\mathrm{ST}}^{[i,i']}$ will tend to 0 as the number of populations tend to $\infty$, but will tend to 0.5 for $F_{\mathrm{ST}}^{[i,i']\mathrm{OS}}$. We believe a value of 0 is more meaningful than 0.5 for these pairs with identical genotypic composition.

Empirically, when comparing $\hat{F}_{\mathrm{ST}}$ and $\hat{F}_{\mathrm{ST}}^{\mathrm{OS}}$ in the 1000 genomes, we see the ranks among chromosomes are not conserved (S3 Table and S4 Fig), because while the transformation between the two estimators is linear, each chromosome will show a different minimum value (and a different arg min value), and hence the linear transformation for each chromosome is different. For example, chromosome 21 is the second lowest with $\hat{F}_{\mathrm{ST}}$ but twelfth lowest with $\hat{F}_{\mathrm{ST}}^{\mathrm{OS}}$. Importantly, we also find that the confidence intervals for $\hat{F}_{\mathrm{ST}}^{\mathrm{OS}}$ are 2.25 to 3.5 times wider than those for $\hat{F}_{\mathrm{ST}}$ (S4 Fig).

## Conclusions

We showed that a moment estimator of $\mathbf{F}_{\mathrm{ST}}$ based on allele-sharing and not making any assumptions about population independence, mating system, ploidy level or inbreeding status of individuals, is unbiased, accurate, fast to calculate and scales easily to genome size data. We provide numerical and analytical solutions for the expectation of $\mathbf{F}_{\mathrm{ST}}$ given migration and mutation rates and population sizes allowing investigators to evaluate how these parameters will affect $\mathbf{F}_{\mathrm{ST}}$. The function `fs.dosage` from the `hierfstat` (0.5-11) R package [37, 38] estimates $\mathbf{F}_{\mathrm{ST}}$ and $\mathbf{F}_{\mathrm{ST_p}}$, as well as individual inbreeding coefficients.

## Supporting information

**S1 Text. A general reference point for kinship and $F_{\mathrm{ST}}$.**
(PDF)

**S2 Text. Allele-sharing, kinship and inbreeding for a $k$−ploid species.**
(PDF)

**S3 Text. Unequal sample sizes and subsampling populations.**
(PDF)

**S4 Text. One thousand genomes population statistics.**
(PDF)

**S1 Fig. Effects of unequal sample sizes and subsampling populations.**
(PDF)

**S2 Fig. Population-specific estimates of $F_{\mathrm{ST}}$ from the 1000 genomes.**
(PDF)

**S3 Fig. RMSEs of $\hat{\mathbf{F}}_{\mathrm{ST}}$ in the 1000 genomes.**
(PDF)

**S4 Fig. Chromosome specific $\hat{F}_{\mathrm{ST}}^{\mathrm{OS}}$ against $\hat{F}_{\mathrm{ST}}$ from the 1000 genomes data.**
(PDF)

**S1 Table. Allele sharing probabilities $A_{jj'}$ between the different possible dosages for ploidies $k$ from 1 to 6.**
(PDF)

**S2 Table. Estimates of chromosome and continent-specific $F_{\mathrm{ST}}$ from the 1000 genomes.**
(PDF)

**S3 Table. Comparison of $\hat{F}_{ST}$ and $\hat{F}_{ST}^{OS}$ in the 1000 genomes data.**
(PDF)

## Acknowledgments

We are grateful to Hugo Corval, Isabela do O, Eleonore Lavanchy and our research groups for discussions about these ideas and comments on previous versions of the manuscript.

## Author Contributions

**Conceptualization:** Jerome Goudet, Bruce S. Weir.

**Data curation:** Jerome Goudet.

**Formal analysis:** Jerome Goudet, Bruce S. Weir.

**Funding acquisition:** Jerome Goudet, Bruce S. Weir.

**Investigation:** Jerome Goudet, Bruce S. Weir.

**Methodology:** Jerome Goudet, Bruce S. Weir.

**Project administration:** Jerome Goudet.

**Resources:** Jerome Goudet.

**Software:** Jerome Goudet.

**Supervision:** Jerome Goudet, Bruce S. Weir.

**Validation:** Jerome Goudet.

**Visualization:** Jerome Goudet.

**Writing – original draft:** Jerome Goudet.

**Writing – review & editing:** Jerome Goudet, Bruce S. Weir.

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
