## [Decision Letter · Decision Letter 0]

14 Sep 2023

Dear Dr Goudet,

Thank you very much for submitting your Methods entitled 'An allele-sharing, moment-based estimator of global, population specific and population pairs  FST under a general model of population structure.' to PLOS Genetics.

The manuscript was fully evaluated at the editorial level and by independent peer reviewers. The reviewers appreciated the attention to an important topic but identified some concerns that we ask you address in a revised manuscript. Particular attention should be paid to the following issues:

As noted by reviewer 1 the role of the Ochoa & Storey (O&S) estimator in the development of the ms is not clear. Generally speaking, Fst's are comparisons of some probabilities of identity relative to some others, and a relevant case in practice is that the "others" are defined from the current, actually observed populations (as in this ms). It appears that O&S's claim that previous estimators are biased rests on applying the general concept in a different way, according to which some ancestral population is considered as a reference. One can debate about the definition of the parameters, as well as about the fairness of assessing bias of estimators relative to a parameter they were not conceived to estimate. The ms restates correctly the definition of the parameters estimated by various estimators, but it also appears to proceed as if the parameter to be estimated should be the one that is easiest to estimate. To address the comments of reviewer 1, the authors should perhaps consider that if the O&S parameter were really the one to estimate, then the results of the ms with respect to estimators would not be so important, and therefore discuss O&S's definition.  

In addition, there are some ambiguities about what exactly is new in this ms. In particular, the introductory review of the literature does not highlight the distinctive features of the estimator discussed here. The estimator uses a general idea of allele sharing and appears to avoid any form of weighting according to per-population sample size. This is distinctly relevant when populations have different expected genetic diversities, so it make sense to apply it to a general migration model (not assuming identical population sizes or migration rates) as done in this ms. It is this combination of features that is new.

The "general population model" is not new per se. Yet, reading the second reviewer's comment, it looks as if it is. This suggests that although various references are cited in the ms after its formulation, the ms could perhaps be more explicit from the start about earlier formulations of the model.

As noted by reviewer 2 the sample sizes in the simulations are not clear, although this is an important consideration in the definition of the estimator (the allele-sharing estimator yields the same estimates as alternative, widely used estimators such as the one of Weir and Cockerham 1984, when the sample sizes are the same among populations).

We therefore ask you to modify the manuscript according to the review recommendations. Your revisions should address the specific points made by each reviewer.

Yours sincerely,

François Rousset

Guest Editor

PLOS Genetics

David Balding

Section Editor

PLOS Genetics

Additional Guest Editor's comments

Miscellaneous:

l.137: I would avoid hasty claims about generality of the equation here as they overlook important issues. For example, with random local extinctions, the form of the recursion may still be correct conditionally on realized extinctions, but if the past history of extinctions is not known, the expected identities and Fst values that may be of interest in any inferential context are expectations over a distribution of past extinction events, and the conditional recursion is not sufficient to compute these expectations.

Mutation is a parameter of model but seems to disappear afterwards. Was it set to zero in Fig. 1, for example?

p.6 l.18-19: I wonder whether the authors really meant to write "correlation" rather than coancestry in this sentence about correlation ... relative to ... correlation. Speaking about correlation here seems to leave something undefined in the reader's mind (namely, correlation of what relative to what?).

Reviewer's **Comments to the Authors:**

Reviewer #1: The authors describe IBD probabilities between all pair of populations under a general model of population structure. They use these IBD probabilities to define matrix-FST, based on the classic definition of F statistics by Rousset (2004). In this matrix-FST, diagonal elements correspond to what it is known as population-specific FST in the literature, and their average correspond to FST. Off-diagonal elements are a somehow new quantity and their interest in conservation genetics is discussed by the authors. Then, the authors present an allele-sharing based estimator of this matrix-FST and they study its statistical properties and compare it to a Bayesian estimator of FST. In addition, the authors discuss another recently proposed estimator of FST based on a different definition of F statistics (Ochoa & Storey, 2022).

I found their work interesting both from a theoretical point of view with their general model for FST, which is a key parameter in population genetics, and from a practical point of view, since computationally efficient estimators are useful with the ever increasing datasets in population genetics. The research is conducted rigorously and conclusions are supported by results. I have not scientific concerns about the originality and validity of this work. However, I think the text needs a thorough revision to improve clarity. The logical connection between different parts of the article needs to be better established. I recommend the text to be revised before it is accepted for publication.

Main concern:

1) The current version of the manuscript does not clearly present the relevance of the analysis of the work of Ochoa & Storey (2022) on FST. In the methods section Ochoa & Storey FST is treated in a brief and separate subsection and results are presented only in the appendix. However, almost half of the discussion is dedicated to it. My impression is that it is not explained to the reader why this particular work is important in the context of their general model and estimator. The text, possibly in the introduction section, needs to be more explicit on what is the role of the Ochoa & Storey FST on the conception, development and results of the rest of work presented.

Minor comments:

2) The section “Statistical properties of FST” (pages 10-11) presents the evaluation of the estimators in a confusing way, in my opinion. I think a general revision would improve the clarity. In particular, I suggest to describe simulations, even if it is briefly (e.g. “a coalescent-with-recombination simulation…”, “simulation of allele frequencies of independent loci…” or other relevant description), before specifying the software used for them.

3) The results section feels like an extended figure caption. I think the reader will follow better the text if the each paragraph starts with a sentence that announces the content of that paragraph rather than the content of a figure. “In the finite island model… (Figure 3)” instead of “Figure 3 shows the results…”.

4) Expected FST in different figures are represented with circles of different colors. These circles are also of different size, and the size seems to be correlated with the absolute FST value. However this is not specified. Please clarify in the caption of the figure to avoid confusion.

5) Use standard conventions for notation: italic letters for variables, including vectors and matrices, and roman letters for labels. In FST, F should be italic (and bold for matrix-FST) and ST in roman. In any case, use an uniform notation, matrix-FST is presented sometimes in italics and sometimes in roman.

6) Paragraph in lines 510-517 uses subindex “W” for FST. If I understood correctly, this subindex needs to be removed to use the same notation that the rest of the article. However, if this has a specific meaning to differentiate to other FST, it needs to be clearly defined in the text.

7) Write acronyms in capital letters for better readability (e.g. IBD instead of ibd).

8) Label panel in figures for better reference.

9) Author summary is missing.

10) Line 117 word “from” duplicated.

Reviewer #2: I have read J. Goudet’s and B. Weir’s manuscript with an immense interest and I must confess, quite an admiration. I have unfortunately not been able to take enough time to review the mathematical aspects of the proposed estimator so my apologies for any lack of comments on this aspect.

After a brief review of FST, including the population-specific definition and their estimators, the authors provide the general expectations for the recurrence of coancestry values between any pair of individuals provided a matrix of migration, thus allowing the calculation for an arbitrary population model. They then propose a moment-estimator for FST and the population-pair-specific FST_P based on the allele sharing between pairs of individuals. Using simulations in population models of increasing complexity, including a river system with two tributaries, they investigate the performance of their estimators and its sensitivity to variable SNP sampling and sample sizes. Lastly, they apply their estimator to the human 1KG dataset and compare the expectations to the ones of another moment-based estimator using a different scaling for the reference population.

I have found the manuscript of excellent quality. The performance analyses are convincing and the discussion thorough and thoughtful. The authors’ estimator has a much higher accuracy than Bayesian estimates in all situations. Its performance is also impressive even for low data quality. I have only very minor comments, and a general call for making the most technical parts slightly more accessible. Congratulations for this work, and looking forward to using this new estimator. Also, I wanted to stress that the relevance of this estimator for conservation genomics, as explained in the Discussion, is of immense interest, since it can quickly provide a set of hypotheses for further modeling analyses. This can be very useful for data exploration and could as well serve as an unbiased summary statistics for ABC studies?

___Things I would be interested to know more about

- One of the advantage of the estimator is its performance even for low sampling size and SNP density. How would it perform for aDNA-style data, including pseudodiploidy? Would it be possible to report a RMSE in such case? I think this would be very useful to show the usability of this method for ancient DNA research.

- Ochoa & Storey’s F_ST^OS estimator uses the minimum AS as a reference point, instead of your estimator using the average. For me, the minimum would lead to a very strong sensitivity to the population sampling design. For instance, suffice that one misses the global population minimum by picking a local one, this would bias the estimate. Instead, what would the results be if using a statistics like the percentile at 10%? A bit on the same topic, you use the average between population, but this could equally be sensitive to the sampling design and to population outliers. Would it be possible to give an intuition, or even, a simulation case using a heterogeneous sampling design with a more robust quantity (e.g. median)?

- Regarding the sampling design, I may have missed something, but it seems that you consider nearly all the populations in your simulated examples? If this is so indeed, could you perform one small performance study in which you consider only a subset of the populations, with a uniform sampling and with a heterogeneous one? e.g. a bit akin to what we have with the 1KG.

- I did not clearly understand why the sensitivity to sampling size would be higher for admixed populations (e.g. Puerto Ricans in the 1KG)?

___For clarification (or confirmation)

- I got a bit confused by the RMSE calculation across the simulation and the application (1KG) cases: what is the reference value that you compare with?

___General

- In general, figures lack resolutions and some were hard to read (labels). Also, the numbers of the scale overlap the color scale, making the “-” sign invisible. On the `FST vs. E[FST]` plots, would be nice to report the CIs.

- I found the description of the methods lacking a bit details, for instance it would be nice to have the commands (if not a full script) for the simulations, including with msprime, as well as for the analyses using fs.dosage().

- In addition, I am lacking info about the methodological analysis of the 1KG dataset. Was there any filtering applied to the SNP set? Was the raw data directly used? If so, or if not, please clarify.

___Specific comments

- L33 A quick conceptual definition of what population-specific FST measures would be nice

- L49 The “F_ST’s” notation is a bit confusing? “’s” is for plural? I am not sure it is necessary.

- L111 Can you be more specific, what “migration rates” do you refer to in this case? I understood that the migration rates you consider afterwards are the intrinsic probabilities.

- L117 Remove one “from”

- L122 Clarify the direction of time: forward or backward, also clarify for the simulators

- Eq2 Would be nice to conceptually decompose the equation

- Fig1 Correct: “sends”, “receives” | Change colors of the lines according to N | Why m in stepping-stone is 1e-2 and not 1e-3 like the others? | Define the x-axis (possibly in legend): generations from what? | Why showing the absolute coancestry and not the actual value?

- L212 Define “r”

- L286 Correct: “are entered”

- L336: What is “c” here? Between the ends of the chromosome? The unit should be number of crossing-overs.

- L342 Correct: “expected”

- L398 I would just say “more heterozygous”, I do not know up to what % you consider it to be highly. The difference is significant but not extreme with some other non-African pops.

- L412 Sorry I may have missed it, but how were loci subsampled? Uniform according to physical position?

- Fig 6 Report in the caption the total number of SNPs from the original dataset (what provides the reference value)

- L462-468 The logic of the paragraph is not crystal-clear, could you develop it or clarify it?

- L470 What do you mean by “the size”? Level?

___Code

I tested the code (hierfstat) package and could run the functions smoothly to estimate FST using simulated individuals. Documentation is present and complete.

Reviewer #3: The paper provides a well-written and constructed overview of recent papers based on FST, and provides some useful results on the accuracy of their moment-based estimator, first described in Weir and Goudet (2017). The authors compare and contrast their approach with two recent methods described in PLoS Genetics. With three such recent papers in PLoS Genetics, FST is clearly well-served! I just have some relatively minor points:

Eqn 2. The need for \\mu in this derivation. It's probably not the best place to put it, but somewhere in the ms, a strength of which is its review of previous work on FST, I would like to see some mention of Slatkin's 1991 definition of FST (at least, as reformulated in Rousset's 1996 Genetics paper, defined in terms of (t_b - t_w)/t_b rather than the GST-way that Slatkin used [no need to mention msats!]). The advantage of Slatkin's definition is that it is then purely genealogical, and could e.g. be applied as a predictive quantity from some coalescent/likelihood/Bayesian algorithm to infer demographic history. Otherwise one tends to have to brush the mutation rate under the carpet. Maybe it is OK for individual SNPs, but for e.g. haplotype-based FSTs one then needs to special-plead the different values you get for different marker types.

333-338. I'm a little unclear on this description. So this is a copy of what they did with sim.genot.metapop.t but with msprime? So presumably something needs to be said about mutation rate as well as map length? Also '20 chromosomes' refers to sample size?

Results: The AFM method clearly does not work very well. Yet the authors do little to unpick what the problem is. Maybe, as the authors note, the computational demands of the method make it not worth it. On the other hand, maybe some discussion of this would be helpful epistemologically. Some lines of thought... Maybe the admixture model assumed in AFM is not very good? The authors note that Gaggiotti and Foll showed satisfactory performance of their likelihood method for the island model, so perhaps their method could be applied in addition to AFM for the island case (Fig 3) to see whether it is an issue intrinsic to the model. To some extent, there is always a bit of an issue to working out the optimal point estimator from a likelihood-based method because the mle is often biased. Maybe try to separate this out in the RMSE? It looks as though RAFM uses MCMC, so is the run-length/convergence an issue?

**Have all data underlying the figures and results presented in the manuscript been provided?**

Reviewer #1: **No: **Current version of the manuscript does not seem to comply with the standards required by PLoS Genetics. Simulated data generated to evaluate the method that represent the main body of results of the article is not made available. The authors should provide the code to generate the simulated data used in this work.

Reviewer #2: Yes

Reviewer #3: Yes

PLOS authors have the option to publish the peer review history of their article (what does this mean?). If published, this will include your full peer review and any attached files.

Reviewer #1: **Yes: **Miguel de Navascués

Reviewer #2: No

Reviewer #3: No

---

## [Editor Report · Decision Letter 1]

31 Oct 2023

Dear Dr Goudet,

We are pleased to inform you that your manuscript entitled "An allele-sharing, moment-based estimator of global, population-specific and population-pair  FST under a general model of population structure." has been editorially accepted for publication in PLOS Genetics. Congratulations!

Yours sincerely,

François Rousset

Guest Editor

PLOS Genetics

David Balding

Section Editor

PLOS Genetics

Comments from the guest editor:

The authors have satisfactorily revised the ms.

**Data Deposition**

http://datadryad.org/submit?journalID=pgenetics&manu=PGENETICS-D-23-00803R1

**Press Queries**

---

## [Editor Report · Acceptance letter]

15 Nov 2023

PGENETICS-D-23-00803R1 

An allele-sharing, moment-based estimator of global, population-specific and population-pair  FST under a general model of population structure. 

Dear Dr Goudet, 

We are pleased to inform you that your manuscript entitled "An allele-sharing, moment-based estimator of global, population-specific and population-pair  FST under a general model of population structure." has been formally accepted for publication in PLOS Genetics! Your manuscript is now with our production department and you will be notified of the publication date in due course.

With kind regards,

Anita Estes

PLOS Genetics

On behalf of:
